# DON'T BE PICKY, ALL STUDENTS IN THE RIGHT FAMILY CAN LEARN FROM GOOD TEACHERS

## ABSTRACT

State-of-the-art results in deep learning have been improving steadily, in good part due to the use of larger models. However, widespread use is constrained by device hardware limitations, resulting in a substantial performance gap between state-of-the-art models and those that can be effectively deployed on small devices.

While Knowledge Distillation (KD) theoretically enables small *student* models to emulate larger *teacher* models, in practice selecting a good student architecture requires considerable human expertise. Neural Architecture Search (NAS) appears as a natural solution to this problem but most approaches can be inefficient, as most of the computation is spent comparing architectures *sampled from the same distribution*, with negligible differences in performance.

In this paper, we propose to instead search for a *family* of student architectures sharing the property of being good at learning from a given teacher. Our approach AutoKD, powered by Bayesian Optimization, explores a flexible graph-based search space, enabling us to automatically learn the optimal student architecture distribution and KD parameters, while being $20\times$ more sample efficient compared to existing state-of-the-art. We evaluate our method on 3 datasets; on large images specifically, we reach the teacher performance while using $3\times$ less memory and $10\times$ less parameters. Finally, while AutoKD uses the traditional KD loss, it outperforms more advanced KD variants using hand-designed students.

## 1 INTRODUCTION

Recently-developed deep learning models have achieved remarkable performance in a variety of tasks. However, breakthroughs leading to state-of-the-art (SOTA) results often rely on very large models: GPipe, Big Transfer and GPT-3 use 556 million, 928 million and 175 billion parameters, respectively (Huang et al., 2019; Kolesnikov et al., 2020; Brown et al., 2020).

Deploying these models on user devices (e.g. smartphones) is currently impractical as they require large amounts of memory and computation; and even when large devices are an option (e.g. GPU clusters), the cost of large-scale deployment (e.g. continual inference) can be very high (Cheng et al., 2017). Additionally, target hardware does not always natively or efficiently support all operations used by SOTA architectures. The applicability of these architectures is, therefore, severely limited, and workarounds using smaller or simplified models lead to a *performance gap* between the technology available at the frontier of deep learning research and that usable in industry applications.

In order to bridge this gap, Knowledge Distillation (KD) emerges as a potential solution, allowing small *student* models to learn from, and emulate the performance of, large *teacher* models (Hinton et al., 2015a). The student model can be constrained in its size and type of operations used, so that it will satisfy the requirements of the target computational environment. Unfortunately, successfully achieving this in practice is extremely challenging, requiring extensive human expertise. For example, while we know that the architecture of the student is important for distillation (Liu et al., 2019b), it remains unclear how to design the optimal network given some hardware constraints.

With Neural Architecture Search (NAS) it is possible to discover an optimal student architecture. NAS automates the choice of neural network architecture for a specific task and dataset, given a *search space* of architectures and a *search strategy* to navigate that space (Pham et al., 2018; Real et al., 2017; Liu et al., 2019a; Carlucci et al., 2019; Zela et al., 2018; Ru et al., 2020). One im-

portant limitation of most NAS approaches is that the search space is very restricted, with a high proportion of resources spent on evaluating very similar architectures, thus rendering the approach limited in its effectiveness (Yang et al., 2020). This is because traditional NAS approaches have no tools for distinguishing between architectures that are similar and architectures that are very different; as a consequence, computational resources are needed to compare even insignificant changes in the model. Conversely, properly exploring a large space requires huge computational resources: for example, recent work by Liu et al. (2019b) investigating how to find the optimal student requires evaluating $10,000$ models. By focusing on the comparison between distributions we ensure to use computational resources only on meaningful differences, thus performing significantly more efficiently: we evaluate $33\times$ less architectures than the most related work to ours (Liu et al., 2019b).

To overcome these limitations, we propose an automated approach to knowledge distillation, in which we look for a *family* of good students rather than a specific model. We find that even though our method, AutoKD, does not output one specific architecture, all architectures sampled from the optimal *family* of students perform well when trained with KD. This reformulation of the NAS problem provides a more expressive search space containing very diverse architectures, thus increasing the effectiveness of the search procedure in finding good student networks.

Our contributions are as follows: (**A**) a framework for combining KD with NAS and effectively emulate large models while using a fraction of the memory and of the parameters; (**B**) By searching for an optimal student family, rather than for specific architectures, our algorithm is up to 20x more sample efficient than alternative NAS-based KD solutions; (**C**) We significantly outperform advanced KD methods on a benchmark of vision datasets, despite using the traditional KD loss, showcasing the efficacy of our found students.

## 2 RELATED WORK

Model compression has been studied since the beginning of the machine learning era, with multiple solutions being proposed (Choudhary et al., 2020; Cheng et al., 2017). Pruning based methods allow the removal of non-essential parameters from the model, with little-to-none drop in final performance. The primary motive of these approaches was to reduce the storage requirement, but they can also be used to speed up the model (LeCun et al., 1990; Han et al., 2015; Li et al., 2016a). The idea behind quantization methods is to reduce the number of bits used to represent the weights and the activations in a model; depending on the specific implementation this can lead to reduced storage, reduced memory consumption and a general speed-up of the network (Fiesler et al., 1990; Soudry et al., 2014; Rastegari et al., 2016; Zhu et al., 2016). In low rank factorization approaches, a given weight matrix is decomposed into the product of smaller ones, for example using singular value decomposition. When applied to fully connected layers this leads to reduced storage, while when applied to convolutional filters, it leads to faster inference (Choudhary et al., 2020).

All the above mentioned techniques can successfully reduce the complexity of a given model, but are not designed to substitute specific operations. For example, specialized hardware devices might only support a small subset of all the operations offered by modern deep learning frameworks. In Knowledge Distillation approaches, a large model (the teacher) distills its knowledge into a smaller student architecture (Hinton et al., 2015b). This knowledge is assumed to be represented in the neural network's output distribution, hence in the standard KD framework, the output distribution of a student's network is optimized to match the teacher's output distribution for all the training data (Yun et al., 2020; Ahn et al., 2019; Yuan et al., 2020; Tian et al., 2020; Tung & Mori, 2019).

The work of Liu et al. (2019b) shows that the architecture of a student network is a contributing factor in its ability to learn from a given teacher. The authors propose combining KD with a traditional NAS pipeline, based on Reinforcement Learning, to find the optimal student. While this setup leads to good results, it does so at a huge computational cost, requiring over 5 days on 200 TPUs. Similarly, Gu & Tresp (2020) also look for the optimal student architecture, but do so by searching for a subgraph of the original teacher; therefore, it cannot be used to substitute unsupported operations.

Orthogonal approaches, looking at how KD can improve NAS, are explored by Trofimov et al. (2020) and Li et al. (2020). The first establishes that KD improves the correlation between different budgets in multi-fidelity methods, while the second uses the teacher supervision to search the architecture in a blockwise fashion.

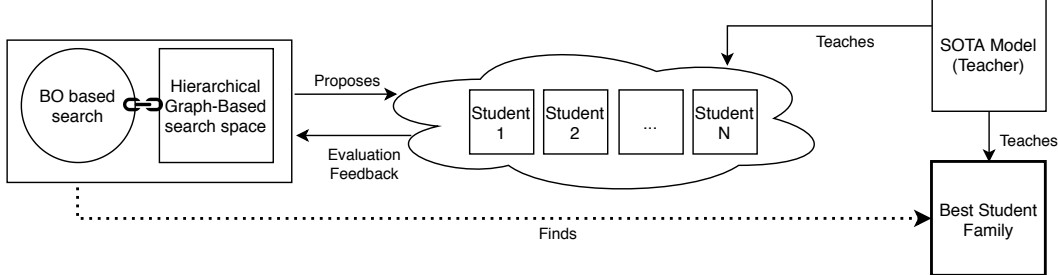

Figure 1: AutoKD leverages multi-fidelity Bayesian Optimization, a hierarchical graph-based search space coupled with an architecture generator optimization pipeline, to find the optimal student for knowledge distillation. See section 3 for a detailed description.

## 3 SEARCHING FOR THE OPTIMAL STUDENT NETWORK GENERATOR

The AutoKD framework (Fig. 1) combines Bayesian Optimization (BO), Neural Architecture Search (NAS) and Knowledge Distillation (KD). AutoKD defines a family of random network generators $G(\boldsymbol{\theta})$ parameterized by a hyperparameter $\boldsymbol{\theta}$, from where student networks are sampled. BO uses a surrogate model to propose generator hyperparameters, while students from these generators are trained with KD using a state-of-the-art teacher network. The student performances are evaluated and provided as feedback to update the BO surrogate model. To improve our BO surrogate model, the *search* procedure is iterated, until the best family of student networks $G(\boldsymbol{\theta}^*)$ is selected. In this section we specify all components of AutoKD. See also Fig. 1 and Algorithm 1 for an overview.

### 3.1 KNOWLEDGE DISTILLATION

Knowledge Distillation (KD; Hinton et al., 2015b) is a method to transfer, or distill, knowledge from one model to another—usually from a large model to small one—such that the small *student* model learns to emulate the performance of the large *teacher* model. KD can be formalized as minimizing the objective function:

$$\mathcal{L}_{\mathrm{KD}} = \sum_{x_i \in X} l(f_T(x_i), f_S(x_i)) \tag{1}$$

where $l$ is the loss function that measures the difference in performance between the teacher $f_T$ and the student $f_S$, $x_i$ is the $i$th input, $y_i$ is the $i$th target. The conventional loss function $l$ used in practice is a linear combination of the traditional cross entropy loss $L_{\mathrm{CE}}$ and the Kullback–Leibler divergence $L_{\mathrm{KL}}$ of the pre-softmax outputs for $f_T$ and $f_S$:

$$l = (1 - \alpha)L_{\mathrm{CE}} + \alpha L_{\mathrm{KL}} \left( \sigma\left(f_T(x_i)/\tau\right), \sigma\left(f_S(x_i)/\tau\right) \right) \tag{2}$$

where $\sigma$ is the softmax function $\sigma(x) = 1/(1 + \exp(-x))$, and $\tau$ is the softmax temperature. Hinton et al. (2015b) propose "softening" the probabilities using temperature scaling with $\tau \geq 1$. The parameter $\alpha$ represents the weight trade-off between the KL loss and the cross entropy loss $L_{\mathrm{CE}}$. The $\mathcal{L}_{\mathrm{KD}}$ loss is characterized by the hyper-parameters: $\alpha$ and $\tau$; popular choices are $\tau \in \{3, 4, 5\}$ and $\alpha = 0.9$ (Huang & Wang, 2017; Zagoruyko & Komodakis, 2016; Zhu et al., 2018). Numerous other methods (Polino et al., 2018; Huang & Wang, 2017; Tung & Mori, 2019) can be formulated as a form of Equation (2), but in this paper we use the conventional loss function $l$.

Traditionally in KD, both the teacher and the student network have predefined architectures. In contrast, AutoKD defines a search space of student network architectures and finds the optimal student by leveraging neural architecture search, as detailed below.

### 3.2 STUDENT SEARCH VIA GENERATOR OPTIMIZATION

Most NAS method for vision tasks employ a cell-based search space, where networks are built by stacking building blocks (*cells*) and the operations inside the cell are searched (Pham et al., 2018; Real et al., 2017; Liu et al., 2019a). This results in a single architecture being output by the NAS procedure. In contrast, more flexible search spaces have recently been proposed that are based on

---

**Algorithm 1:** AutoKD

---

1: **Input:** Network generator $G$, BOHB hyperparameters($\eta$, training budget $b_{min}$ and $b_{max}$), Evaluation function $f_{KD}(\boldsymbol{\theta}, b)$ which assesses the validation performance of a generator hyperparameter$\boldsymbol{\theta}$ by sampling an architecture from $G(\boldsymbol{\theta})$ and training it with the KD loss $\mathcal{L}_{KD}$ (equations 1 and 2) for $b$ epochs.

2: $s_{max} = \lfloor \log_\eta \frac{b_{max}}{b_{min}} \rfloor$;

3: **for** $s \in \{s_{max}, s_{max} - 1, \ldots, 0\}$ **do**

4:     Sample $M = \lceil \frac{s_{max}+1}{s+1} \cdot \eta^s \rceil$ generator hyperparameters $\boldsymbol{\Theta} = \{\boldsymbol{\theta}^j\}_{j=1}^M$ which maximises the raito of kernel density estimators ;    ▷ (Falkner et al., 2018, Algorithm 2)

5:     Initialise $b = \eta^s \cdot b_{max}$ ;    ▷ Run Successive Halving (Li et al., 2016b)

6:     **while** $b \leq b_{max}$ **do**

7:         $\mathbf{L} = \{f_{KD}(\boldsymbol{\theta}, b) : \boldsymbol{\theta} \in \boldsymbol{\Theta}\}$;

8:         $\boldsymbol{\Theta} = top\_k(\boldsymbol{\Theta}, \mathbf{L}, \lfloor |\boldsymbol{\Theta}|/\eta \rfloor)$;

9:         $b = \eta \cdot b$;

10:     **end while**

11: **end for**

12: Obtain the best performing configuration $\boldsymbol{\theta}^*$ for the student network generator.

13: Sample $k$ architectures from $G(\boldsymbol{\theta}^*)$, train them to completion, and obtain test performance.

---

neural network generators (Xie et al., 2019; Ru et al., 2020). The generator hyperparameters define the characteristics of the family of networks being generated.

NAGO optimizes an architecture generator instead of a single architecture and proposes a hierarchical graph-based space which is highly expressive yet low-dimensional (Ru et al., 2020). Specifically, the search space of NAGO comprises three levels of graphs (where the node in the higher level is a lower-level graph). The top level is a graph of cells ($G_{top}$) and each cell is itself a graph of middle-level modules ($G_{mid}$). Each module further corresponds to a graph of bottom-level operation units ($G_{bottom}$) such as a relu-conv3×3-bn triplet. NAGO adopts three random graph generators to define the connectivity/topology of $G_{top}$, $G_{mid}$ and $G_{bottom}$ respectively, and thus is able to produce a wide variety of architectures with only a few generator hyperparameters. AutoKD employs NAGO as the NAS backbone for finding the optimal student family.

Our pipeline consists of two phases. In the first phase (*search*), a multi-fidelity Bayesian optimisation technique, BOHB (Falkner et al., 2018), is employed to optimise the low-dimensional search space. BOHB uses partial evaluations with smaller-than-full budget to exclude bad configurations early in the search process, thus saving resources to evaluate more promising configurations. Given the same time constraint, BOHB evaluates many more configurations than conventional BO which evaluates all configurations with full budget. As Ru et al. (2020) empirically observe that good generator hyperparameters lead to a tight distribution of well-performing architectures (small performance standard deviation), we similarly assess the performance of a particular generator hyperparameter value with only one architecture sample. In the second phase (*retrain*A), AutoKD uniformly samples multiple architectures from the optimal generator found during the search phase and evaluates them with longer training budgets to obtain the best architecture performance.

Instead of the traditionally used cross-entropy loss, AutoKD uses the KD loss in equation 2 to allow the sampled architecture to distill knowledge from its teacher. The KD hyperparameters temperature $\tau$ and loss weight $\alpha$ are included in the search space and optimized simultaneously with the architecture to ensure that the student architectures can efficiently distill knowledge both from the designated teacher and the data distribution. A full overview of the framework is shown in Fig. 1.

## 4 EXPERIMENTS

The first part of this section studies how KD can improve the performance of our chosen NAS backbone (NAGO). In the second part, we show how a family of students, when trained with KD (AutoKD), can emulate much larger teachers, significantly outperforming current hand-crafted architectures.

**Experimental setup.** All of our experiments were run on the two, small-image, standard object recognition datasets *CIFAR10* and *CIFAR100* (Krizhevsky, 2009), as well as *MIT67* for large-image scene recognition (Quattoni & Torralba, 2009). We limit the number of student network parameters to 4.0M for small-image tasks and 6.0M for large-image tasks. Following Liu et al. (2019b), we picked Inception-Resnet-V2 (Szegedy et al., 2016) as a teacher for the large image dataset. As that model could not be directly applied to small images, and to explore the use of a machine-designed network as a teacher, we decided to use the best DARTS (Liu et al., 2019a) architecture to guide the search on the CIFAR datasets. For *ImageNet* (Deng et al., 2009), we use a Inception-Resnet-V2 teacher. All experiments are run on NVIDIA Tesla V100 GPUs.

**NAS implementation.** Our approach follows the search space and BO-based search protocol proposed by NAGO (Ru et al., 2020), as such our student architectures are based on hierarchical random graphs. Likewise, we employ a multi-fidelity evaluation scheme based on BOHB (Falkner et al., 2018) where candidates are trained for different epochs (30, 60 and 120) and then evaluated on the validation set. In total, only ∼300 models are trained during the search procedure: using 8 GPUs, this amounts to ∼2.5 days of compute on the considered datasets. At the end of the search, we sample 8 architectures from the best found generator, train them for 600 epochs (with KD, using the optimal temperature and loss weight found during the search), and report the average performance (top-1 test accuracy). All remaining training parameters were set following Ru et al. (2020).

In AutoKD, we include the knowledge distillation hyperparameters, *temperature* and *weight*, in the search space, so that they are optimized alongside the architecture. The temperature ranges from 1 to 10, while the weight ranges from 0 to 1. Fig. 8 (Appendix) illustrates the importance of these hyperparameters when training a randomly sampled model, lending support to their inclusion.

## 4.1 IMPACT OF KNOWLEDGE DISTILLATION ON NAS

To understand the contribution from KD, we first compare vanilla NAGO with AutoKD on CIFAR100. Fig. 2 shows the validation accuracy distribution at different epochs: clearly, using KD leads to better performing models. Indeed this can be seen in more detail in Fig. 3, where we show the performance of the best found model vs the wall clock time for each budget. It is worth mentioning that while the KD version takes longer (as it needs to compute the lessons on the fly), it consistently outperforms vanilla NAGO by a significant margin on all three datasets.

Note that accuracies in Fig. 3 refer to the best models found during the search process, while Fig. 2 shows the histograms of all models evaluated during search, which are by definition lower in accuracy, on average. At the end of search, the model is retrained for longer (as commonly done in NAS methods), thus leading to the higher accuracies also shown in Figs. 6, 7.

Not only does AutoKD offer better absolute performance, but it also enables better multi-fidelity correlation, as can be seen in Fig. 4. For example, the correlation between 30 and 120 epochs improves from 0.49 to 0.82 by using KD, a result that is consistent with the findings in Trofimov et al. (2020). Note that multi-fidelity methods work under the assumption that the rankings at different budgets remains consistent to guarantee that the best models progress to the next stage. A high correlation between the rankings is, as such, crucial.

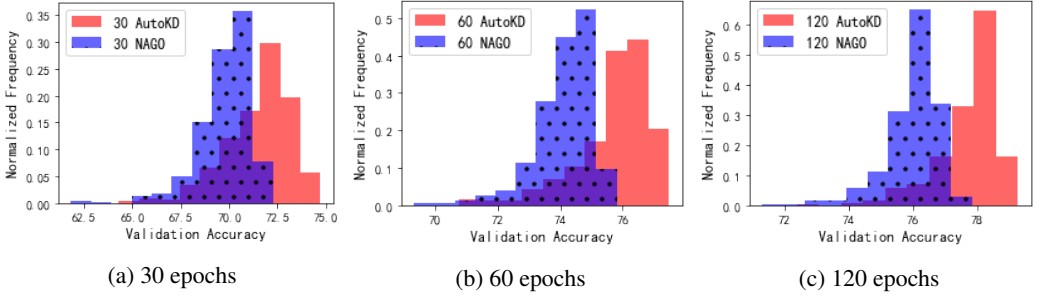

(a) 30 epochs      (b) 60 epochs      (c) 120 epochs

Figure 2: Top-1 accuracy distribution for AutoKD and standard NAGO at different budgets on CIFAR100. The histograms are tallied across 5 runs. Across all budgets, AutoKD samples architectures with improved performances in top-1 accuracy compared to NAGO.

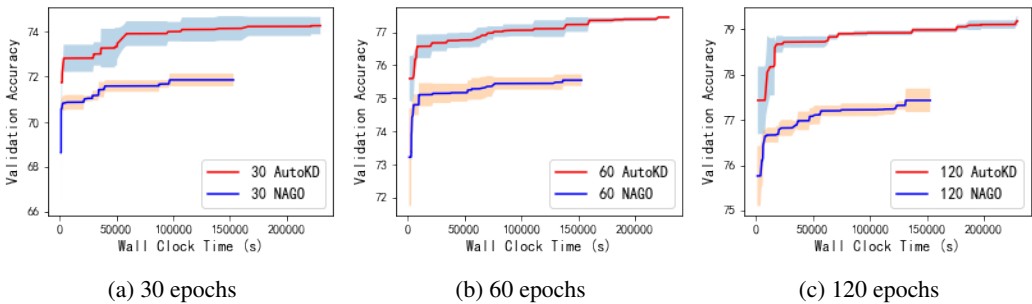

| (a) 30 epochs | (b) 60 epochs | (c) 120 epochs |

Figure 3: Top-1 accuracy of the best model found during search at a given computation time on CIFAR100 for AutoKD (red) and NAGO (blue) across different budgets. Each method was run 8 times with the bold curve showing the average performance and the shaded region the stdev.

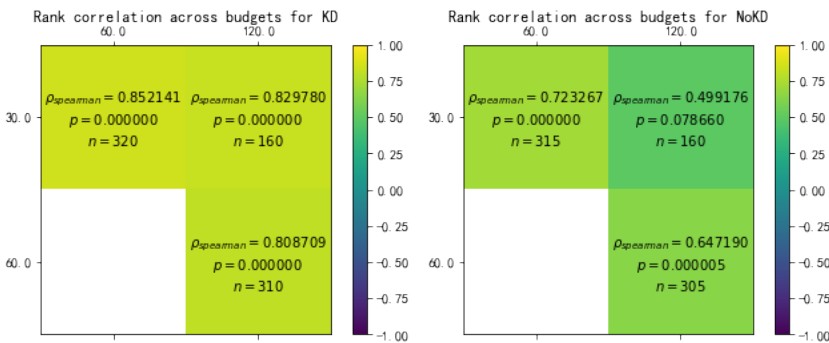

Figure 4: Rank Correlations between different epoch budgets for AutoKD (KD; left) and Standard NAGO (right) computed for 5 runs of NAGO and AutoKD respectively. NAGO reports a rank correlation coefficient of $5 \cdot 10^{-1}$ for epoch pair 30-120, which is $3.3 \cdot 10^{-1}$ less than that of the KD rank correlation. These results show that the rank correlation across all budget pairs vastly improves when knowledge distillation is applied.

## 4.2 LARGE MODEL EMULATION

At its core, AutoKD's goal is to emulate the performance of large SOTA models with smaller students. Fig. 6 shows how the proposed method manages to reach the teacher's performance while using only 1/9th of the memory on small image datasets. On MIT67, the found architecture is not only using 1/3rd of the memory, but also 1/10th of parameters. Finally, it is worth noting how AutoKD increases student performance, as such the high final accuracy cannot only be explained by the NAS procedure. Indeed, looking at Fig. 7 it is clear how KD improves both the speed of convergence and the final accuracy. Furthermore, as shown in Fig. 5, the optimal family of architectures is actually different when searched with KD.

*MIT67, CIFAR100, CIFAR10.* Table 1 shows the comparison of AutoKD with other KD methods. Notice how learning the student architecture allows AutoKD to outperform a variety of more advanced KD approaches while emplying a smaller parameter count in the student. The exception to this is CIFAR10, where AutoKD outperforms other methods but with a larger number of parameters. This is because the default networks in the NAGO search space have 4M parameters, which is too large for this application. Accuracy-wise, the only method doing better on CIFAR100, Yuan et al. (2020), does so with a student with significantly more parameters (34M vs 4M). Finally, AutoKD is orthogonal to advanced KD approaches and could be combined with any of them for even further increases in performance.

*ImageNet.* The improved results on smaller datasets extend to large datasets as well. On ImageNet, AutoKD reaches 78.0% top-1 accuracy, outperforming both Liu et al. (2019b) using the same teacher (75.5%) and vanilla NAGO (76.8%).

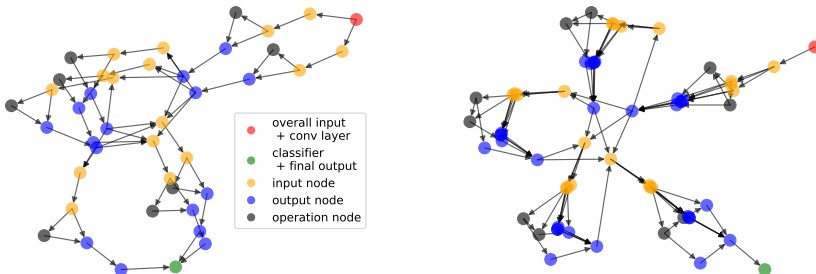

Figure 5: Networks sampled from the best generator parameters found by AutoKD (left) and NAGO (right) on CIFAR10.The former is organized with 8 clusters of 3 nodes, while the latter has 5 clusters of 10 nodes, showcasing how the optimal configuration depends on teacher supervision. Arrows indicate information flow between nodes. As NAGO's search space is hierarchical, it contains a number of sub-graphs; the yellow (blue) nodes are the input (output) nodes of these sub-graphs while the grey nodes are the operation units (conv-norm-relu).

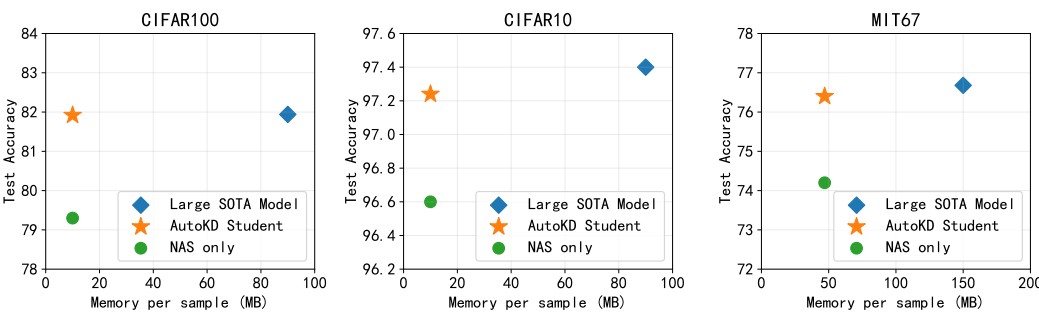

Figure 6: Accuracy vs memory per sample for the SOTA Model (the teacher), the AutoKD student and best architecture found by vanilla NAS. This plot clearly shows how AutoKD finds a model superior to NAS-only, managing to reach the performance of the large teacher model while using a fraction of the per sample memory. Note that the MIT67 teacher has almost $10\times$ the number of parameters of the student (54M vs 6M).

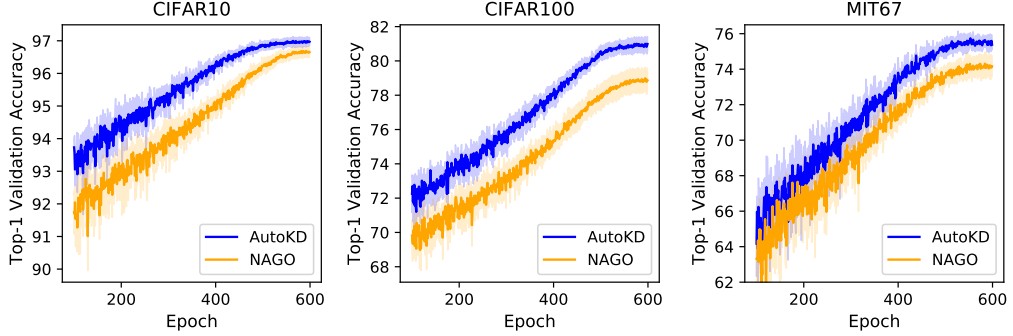

Figure 7: Final training curves for the top generator found by NAGO and AutoKD, for CIFAR10, CIFAR100 and MIT67. Each generator was sampled 8 times and the 8 corresponding architectures trained for 600 epochs. Bold line represents the average; shaded region represents std deviation.

## 5 DISCUSSION AND CONCLUSION

Improving Knowledge Distillation by searching for the optimal student architecture is a promising idea that has recently started to gain attention in the community (Liu et al., 2019b; Trofimov et al., 2020; Gu & Tresp, 2020). In contrast with earlier KD-NAS approaches, which search for specific architectures, our method searches for a *family* of networks sharing the same characteristics. The

Table 1: Comparison with KD state-of-the-art. AutoKD uses the standard KD loss (Hinton et al., 2015b), while competing methods are using modern variants. Improvement of student accuracy in parenthesis is with respect to the same student without KD. "↑×$f$" and "↓×$f$" denotes the increase/decrease in parameter count by the factor $f$ relative to AutoKD for the same dataset. The top performing student accuracy ($S\ acc$) for each dataset is specified in bold. For each dataset, we sampled 8 architectures and averaged them over 5 runs. For MIT67, the VID method is a transfer learning task from ImageNet to MIT67, hence the absence of teacher accuracy ($T\ acc$) statistics.

| Method† | Teacher (T) | Student (S) | S params | T acc | S acc |
|---|---|---|---|---|---|
| **MIT67** | | | | | |
| **SKD** | ResNet-18 | ResNet-18 | 11.5M ↑×1.9 | 55.3 | 60.4 (+5.1) |
| **VID** | ResNet-34 | ResNet-18 | 11.5M ↑×1.9 | — | 71.9 (+0.9) |
| **VID** | ResNet-34 | VGG-9 | 10.9M ↑×1.8 | — | 72.0 (+6.0) |
| **AutoKD** (ours) | InceptionResNetV2 | NAGO | 6.0M | 76.6 | **76.0** (+1.8) |
| **CIFAR100** | | | | | |
| **CRD** | ResNet-32 × 4 | ShuffleNetV2 | 7.4M ↑×1.9 | 79.4 | 75.7 (+3.8) |
| **CRD** | WRN 40-2 | WRN-16-2 | 0.7M ↓×5.7 | 75.6 | 75.6 (+2.4) |
| **VID** | WRN 40-2 | WRN 40-2 | 2.2M ↓×1.8 | 74.2 | 76.1 (+1.8) |
| **KD-LSR** | ResNet-18 | ResNet-18 | 11.5M ↑×2.9 | 75.9 | 77.4 (+1.5) |
| **SKD** | ResNet-18 | ResNet-18 | 11.5M ↑×2.9 | 75.3 | 79.6 (+4.3) |
| **KD-LSR** | DenseNet-121 | DenseNet-121 | 7.0M ↑×1.8 | 79.0 | 80.3 (+1.3) |
| **KD-LSR** | ResNeXt29 | ResNeXt29 | 34.2M ↑×8.6 | 81.0 | **82.1** (+1.1) |
| **AutoKD** (ours) | DARTS | NAGO | 4.0M | 81.9 | 81.2 (+2.6) |
| **CIFAR10** | | | | | |
| **VID** | WRN 40-2 | WRN 16-1 | 0.7M ↓×5.7 | 94.3 | 91.9 (+1.2) |
| **SPKD** | WRN 16-8 | WRN 40-2 | 2.2M ↓×1.8 | 95.8 | 95.5 (+0.6) |
| **AT** | WRN 16-8 | WRN 40-2 | 2.2M ↓×1.8 | 95.8 | 95.5 (+0.6) |
| **AutoKD** (ours) | DARTS | NAGO | 4.0M | 97.4 | **97.1** (+0.8) |

† SKD (Yun et al., 2020), VID (Ahn et al., 2019), KD-LSR (Yuan et al., 2020), CRD (Tian et al., 2020), SPKD (Tung & Mori, 2019), AT (Zagoruyko & Komodakis, 2016).

main benefit of this approach is sample efficiency: while traditional methods spend many computational resources evaluating similar architectures (Yang et al., 2020), AutoKD is able to avoid this pitfall: for instance, the method of Liu et al. (2019b) requires ∼ 10, 000 architecture samples, while AutoKD can effectively search for the optimal student family with only 300 samples. Compared to traditional KD methods, AutoKD is capable of achieving better performance with student architectures that have less parameters (and/or use less memory) than hand-defined ones.

Our message "DON'T BE PICKY" refers to the fact that the macro-structure (connectivity and capacity) of a network is more important than its micro-structure (the specific operations). This has been shown to be true for non-KD NAS (Xie et al., 2019; Ru et al., 2020) and is here experimentally confirmed for KD-NAS as well. Changing the focus of optimization in this way releases computational resources that can be used to effectively optimize the global properties of the network. Additionally, the fact that a family of architectures can characterized by a small number of hyperparameters makes the comparison of architectures more meaningful and interpretable. In the current implementation, AutoKD finds the optimal student family, in which all sampled architectures perform well: future work should explore how to fully exploit this distribution, possibly finetuning the network distribution to obtain an ever better performing model.

To summarize, AutoKD offers a strategy to efficiently emulate large, state-of-the-art models with a fraction of the model size. Indeed, our family of searched students consistently outperforms the best hand-crafted students on CIFAR10, CIFAR100 and MIT67.

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

## A   APPENDIX

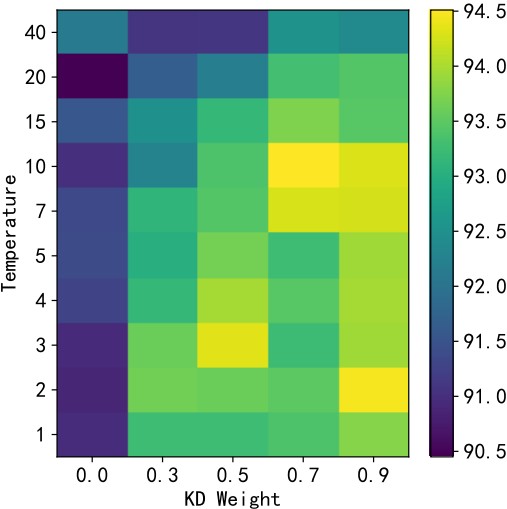

Figure 8: Student model test accuracy for various temperature ($\tau$) and loss weight ($\alpha$) combinations. The model was sampled from a generator with random parameters, and trained with KD on CIFAR10 using the DARTS teacher. The table suggests that there is a positive correlation between the KD loss weight and the performance of the student model. Note that the variability shown when the loss is set to 0 is solely due to the inherent stochasticity of the training process.

