# OpenReview forum: "Don't be picky, all students in the right family can learn from good teachers"
_ICLR.cc/2021/Conference — Reject_

### Official Review · AnonReviewer2 · 2020-10-22
**A very rough method: NAS on KD directly, for what?**

**Rating:** 3
**Confidence:** 4

**Review:**


Summary and contributions
  The paper takes advantage of NAGO and proposes to search for a family of student network architectures instead of a single architecture, aiming to be more sample efficient. This reformulation of the NAS problem makes it possible to search in an expressive searching space, at the same time, avoid to waste time in comparing similar architectures.

Strengths
  This KD-NAS approach further develops the benefit of NAGO and makes it convenient to search for a family of student architectures. The optimization objective of finding a family instead of a single architecture helps to speed up NAS process or more sample efficient.

Weaknesses
  There have already exist KD-NAS approaches and the main difference of this work is to search for a family. This objective mainly takes advantage of the generator in NAGO, so the contribution and novelty should be reduced accordingly.
  The purpose of some experiments in this paper is a little confusing. From my point of view, this paper aims to utilize NAS to benefit KD, at the same time, make NAS more efficient. Maybe you should compare AutoKD with earlier KD-NAS approaches instead of NAGO to show the impact of KD on NAS.
  The title shows the fact that the macro-structure of a network is more important than its micro-structure, which has been studied in previous work. But this fact doesn’t logically lead to idea of ‘searching for a family’. Also, I think it isn’t clearly articulated that how this family of student architectures can benefit knowledge distillation. Maybe the performance gains result from the ensemble of networks.
  Finally, I think it doesn't make much sense to have the comparisons showed in Figure 7. Besides, the logic and results shown in the visualization of Figure 6 are not clear enough to me.


[ Detailed comments]
1. In Chapter 3.2, the original meaning is unclear: ‘The hyperparameters ... that it represents.’
2. In Algorithm 1, there are many unmarked sentence endings. In addition, where are the definitions of functions f and α, and what does D refer to?
3. In Figure 6, the specific meanings of various arrows and various colors need to be marked.

---

> ### Author Response · Authors · 2020-11-20
> **Replies to AnonReviewer2**
>
> **"Maybe you should compare AutoKD with earlier KD-NAS approaches instead of NAGO to show the impact of KD on NAS".**
> As far as we are aware, only two methods exist combining NAS with KD: (a) Trofimov et al (2020) shows that KD improves the accuracy correlation between different fidelity evaluations, which is important in a multi-fidelity approach, but do not report the results of the final model retrained with KD, as it's not their purpose. As such, no comparison is possible. The authors' approach of using KD to improve NAS is also conceptually different from our approach where NAS is used to improve KD.
> (b) Liu et al (2019b) is the work most similar to ours and we are currently running experiments on ImageNet to directly compare with them. Nevertheless, we are 33x more sample efficient, which is a non-trivial improvement.
>
> **"The title shows the fact that the macro-structure of a network is more important than its micro-structure, which has been studied in previous work. But this fact doesn’t logically lead to idea of ‘searching for a family’. Also, I think it isn’t clearly articulated that how this family of student architectures can benefit knowledge distillation".**
> We agree that searching for a family/distribution rather than a specific architecture is not the only possibility to solve those issues, but it is a natural way in that direction, and has been shown to be successful. Specifically, searching for a family instead of a single architecture allows our method to be be extremely efficient compared to traditional NAS approaches.
> See also the reply to your comment about 'ensembles'.
>
> **"Maybe the performance gains result from the ensemble of networks".**
> Our method does not use ensembles of networks. By optimizing the generator, we find the best family of architectures (networks that share similar characteristics according to the generator parameters). Samples from this family are then evaluated. We report the average accuracy of the optimal family, as there is natural variability. This is the average of the accuracies, not of the predictions (as would be in ensembles).
>
> **"it doesn't make much sense to have the comparisons showed in Figure 7".**
> The reviewer does not explain the reasons for this comment, but we agree that Fig.7 was not absolutely clear and would like to clarify it. We have relabeled the (blue, orange, green) points from (Teacher, AutoKD, Student) to (Large SOTA Model, AutoKD-Student, NAS), which is a more accurate description of the 3 networks shown. 'NAS' does not use KD.
>
> The empirical support for the benefits of AutoKD, given in Fig.7, are as follows: (1) compared to standard NAS (green), AutoKD (orange) shows the benefit of training the best student network with knowledge distillation from a fixed teacher network (blue), while maintaining the same memory complexity; (2) compared to a fixed teacher network (blue),  AutoKD (orange) is able to be compressed to achieve a lower memory-complexity with negligible loss in accuracy. The green and orange points, share the same memory complexity by design, as they use the same architecture.
> These details will be added to the Figure for the final version.
>
> **Regarding the [Detailed comments].**
> Thank you for this feedback, we are currently improving the paper.
>
> **Regarding comparison with other KD+NAS methods.**
> Please see our "GENERAL COMMENT" reply.

---

### Official Review · AnonReviewer1 · 2020-10-27
**Limited novelty**

**Rating:** 3
**Confidence:** 3

**Review:**

This paper applied knowledge distillation (KD) on network architecture generator optimization (NAGO) which is one of NAS. Specifically, KD has used in the procedure of searching and the parameter of KD has involved in the search space. As a result, the proposed method (AutoKD) seems to improve NAGO.

Pros)
- Extensive experiments are performed

Cons)
- Some figures are not clearly shown. Please refine the figures (e.g., figure 2) for clarity.
- Applying KD into NAGO seems to be naively done. It seems that the proposed method is incremental and the contribution is limited and the differences are not highlighted.
- The result comparison in Table 1 looks not fair:
  - KD used in this paper used better teachers following the convention, but the competitor KD-LSR and SKD in the table are self-distillation methods, so the comparison is meaningless.
  - On CIFAR100 dataset, CRD in the original paper used WRN-40-2 as a teacher and trained the student of WRN-16-2, which has only 0.7M parameters with an accuracy of 75.64. However, this paper reports CRD used ShuffleNetV1 which have more parameters
  - On MIT67 dataset, VID used an ImageNet-pretrained model for transfer learning, but AutoKD used the fine-tuned teacher which is much beneficial to KD in terms of performance.
  - On CIFAR10 dataset, the compared models (WRN 16-1 and two WRN 40-2s) have fewer parameters than that of NAGO for AutoKD. Therefore, it is hard to say that AutoKD outperforms them.
- Experimental results are somewhat unconvincing:
   - As weight is zero in Figure 2, the accuracies in the table should show consistent performance but are deviated w.r.t temperature. The authors should clarify this.
   - Why the accuracies of NAGO in Figure 4 look low compared to the other results in the paper?
- Using KD on NAS leverages additional computational cost, but it is not clearly compared quantitatively


Comments)
- The method is incremental, and the novelty is limited. The experimental results comparing with other methods are biased to the proposed method, where the competitors' performances are not fairly compared, so it is hardly convincing the results and the effectiveness of the proposed method.

---

> ### Author Response · Authors · 2020-11-21
> **Replies to AnonReviewer1**
>
> **"Figures".**
> We are currently updating the figures to improve their clarity.
>
> **"Result comparison in Table 1".**
> We agree that comparison with some other methods is not on par; in fact, finding completely fair baselines against which to compare AutoKD proved extremely difficult.
> Nonetheless they are competing approaches that we think deserve acknowledgement and have their own strengths. For instance, though using 8.6x more parameters, KD-LSR(ResNeXt29) outperforms AutoKD on CIFAR100, and we believe that readers will find this comparison useful.
> We will happily add further comparison suggested by the reviewer.
> Please also see our "GENERAL COMMENT" reply.
> Regarding  CIFAR100/CRD, we apologize for the typo and will fix it in our next revision.
>
> **"Variability in Figure 2 when weight is zero."**
> The variability in the 0 weight column is simply a function of the natural stochasticity of the training process.
>
> **"Figure 4 accuracies".**
> The accuracies in Figure 4 refer to the *best models* found during the search process, while Figure 3 shows the histograms of *all models* evaluated during search, which are by definition lower in accuracy, on average.
> At the end of search, the model is retrained for longer (as commonly done in NAS methods), thus leading to the higher accuracies shown in Figure 7 and 8.
> We will clarify this is the caption.
>
> **"Computational cost".**
> We explicitly compare with our direct competitor (Liu et al., 2019b), showing to be 33x more sample efficient. It is unclear how to compare with traditional KD approaches in which the student architecture has been optimized by humans in an undisclosed amount of time.

---

> > ### Author Response · Authors · 2020-11-24
> > **Update following AnonReviewer1 feedback**
> >
> > A new version of the paper has been uploaded. The following changes were made, following the reviewer's comments:
> >
> > **"Figures."**
> > Old Figure 2, figure improved and is now Figure 8 in the Appendix.
> > Now Figure 3, caption has been clarified.
> > Now Figure 6, figure and caption have been clarified.
> >
> > **"Result comparison in Table 1".**
> > The typo has been fixed.
> >
> > **"Variability in Figure 2 when weight is zero."**
> > This Figure has been moved to the Appendix (Now Figure 8) and the caption has been updated with an explanation for the variability when weight is zero:
> >
> > "Note that the variability shown when the loss is set to 0 is solely due to the inherent stochasticity of the training process."
> >
> > **"Figure 4 accuracies".**
> > We have added the following to Section 4.1 and clarified the figure captions where appropriate:
> >
> > ... "Note that accuracies in Figure 3 refer to the best models found during the search process, while Figure 2 shows the histograms of all models evaluated during search, which are by definition lower in accuracy, on average. At the end of search, the model is retrained for longer (as commonly done in NAS methods), thus leading to the higher accuracies also shown in Figures 6, 7."

---

### Official Review · AnonReviewer3 · 2020-11-01
**Nice addition for combining architecture search and knowledge distillation, but lacking support for some claims**

**Rating:** 5
**Confidence:** 4

**Review:**

Summary:

This paper proposes searching for an architecture generator that outputs good student architectures for a given teacher. The authors claim that by learning the parameters of the generator instead of relying directly on the search space, it is possible to explore the search space of architectures more effectively, increasing the diversity of the architectures explored. They show that this approach combined with the standard knowledge distillation loss is able to learn good student architectures requiring substantially less samples and achieving competitive performances when comparing to other knowledge distillation algorithms.

Pros:
+ The paper is clear overall. A system for combining knowledge distillation and architecture search is proposed that combines surrogate functions, multi-fidelity optimization, and neural architecture generators.
+ The results on CIFAR10 and CIFAR100 are compelling.

Cons:

- Somewhat limited conceptual innovation. The authors combine NAGO and BOHB to obtain a solution for architecture search for knowledge distillation. While the results are solid, there is little insight about the behavior of the method, e.g., no ablations are performed beyond the comparison with NAGO, so it is hard to assess the importance of the individual components.
- No results on ImageNet.
- The results are compared with models that use other teacher architectures, so it is hard to determine if the improvements are due to an improved teacher or a better knowledge distillation method. The fact that both student architecture size and student accuracy are important metrics means that there is no easy way
- No code is included, but I assume that this is something that the authors will address for the final version.
- The use of BOHB and the surrogate loss are insufficiently described, for example, in Algorithm 1.
- The claim that a standard neural architecture search would produce architectures sampled from the same distributions and therefore not be efficient in exploring the space of students is insufficiently explored.

Comments:

In essence, this paper only corresponds to a difference in how the architectures for knowledge distillation are generated. The claim that not going through the architecture generator would lead to architectures that are too similar lacks substantiation. Additionally, the paper does not have information about comparisons with other architecture search algorithms for knowledge distillation or surrogate functions, therefore the introduction of a new framework may not be warranted under the claims of the authors.

---

> ### Author Response · Authors · 2020-11-21
> **Replies to AnonReviewer3**
>
> **"Ablation studies".**
> AutoKD is a combination of a NAS technique with KD. Other than showing the improvement obtained over the performance of the NAS component alone (which we show in Figures 3,4,5,7 and 8) it is not clear to us what kind of further ablations would make sense. Could the reviewer clarify what he has in mind for ablations? On a related note, Figure 2 shows the beneficial impact of learning KD temperature and weight.
>
> **"ImageNet experiments".**
> We are performing the ImageNet experiments at the moment and will provide the results as soon as they are concluded.
>
> **"Code".**
> Yes, the code will be released.
>
> **"The use of BOHB and the surrogate loss are insufficiently described".**
> A detailed description of BOHB is given in Ru et al (2020) and relevant references therein, but we will provide a more in-depth explanation here as well, and improve the clarify of Algorithm 1 accordingly.
>
> **"The claim that a standard neural architecture search would produce architectures sampled from the same distributions and therefore not be efficient in exploring the space of students is insufficiently explored".**
> We apologize for failing to communicate clearly. Standard NAS approaches have no tools for distinguishing between architectures that are similar and architectures that are very different; as a consequence, computational resources are needed to compare even insignificant changes in the model. By focusing on the comparison between distributions we ensure to use computational resources only on meaningful differences, thus performing significantly more efficiently (We evaluate 33x less architectures than our direct competitor Liu et al, 2019b).
>
> **"The claim that not going through the architecture generator would lead to architectures that are too similar lacks substantiation".**
> These claims are substantiated in the literature. Specifically, non-generator vs. generator-based NAS has been explored in detail in Yang et al (2020) and Ru et al (2020)---the first work shows how 'narrow' traditional NAS search spaces are, and how most computation is spent searching for very similar architectures with tiny variations in performance; and the second work presents a more expressive search space based on generators, which is also cheaper to optimize.
> As a result, we felt that redoing those experiments would not add substantial value to our paper.
>
> We will improve the 3 last points in the final version, thank you for pointing out the lack of clarity.
>
> Regarding novelty and comparison fairness, see our "GENERAL COMMENT" reply.

---

> > ### Author Response · Authors · 2020-11-24
> > **Update following AnonReviewer3 feedback**
> >
> > A new version of the paper has been uploaded. The following changes were made, following the reviewer's comments:
> >
> > **"No results on ImageNet".**
> > We have added ImageNet results to Section 4.2:
> >
> > "ImageNet. The improved results on smaller datasets extend to large datasets as well. On ImageNet, AutoKD reaches78.0% top-1 accuracy, outperforming both Liu et al. (2019b)(75.5%), using the same teacher, and vanilla NAGO (76.8%)."
> >
> >
> > **"The use of BOHB and the surrogate loss are insufficiently described".**
> > Algorithm 1 has been updated with more information.
> >
> >
> > **"The claim that a standard neural architecture search would produce architectures sampled from the same distributions and therefore not be efficient in exploring the space of students is insufficiently explored".**
> > We added the following to the introduction:
> >
> > ... "This [a high proportion of resources spent on evaluating very similar architectures in traditional NAS] is because traditional NAS approaches have no tools for distinguishing between architectures that are similar and architectures that are very different; as a consequence, computational resources are needed to compare even insignificant changes in the model."
> >
> > ... "By focusing on the comparison between distributions we ensure to use computational resources only on meaningful differences, thus performing significantly more efficiently: we evaluate 33$\times$ less architectures than the most related work to ours (Liu et al., 2019b)."

---

### Author Response · Authors · 2020-11-20
**GENERAL COMMENT**

We thank the reviewers for their feedback. It seems we failed to communicate clearly and as a consequence a number of misconceptions arose. We would like to clarify those with this reply.

======  GOAL AND MERITS

Our goal is to transfer the knowledge from powerful SOTA networks to much smaller models for real life deployment. KD offers a solution, but its performance can be constrained by the architecture of the student. We thus propose to optimize the student architecture via NAS.
NAS solutions not based on generators have proved to be extremely costly as they spend huge computational resources comparing similar architectures (see Yang et al, 2020).
In contrast, we argue that once a good enough distribution (family) of architectures is found, any of them will perform well (see Ru et al, 2020).
This distinction enables us to be 33x more sample efficient, reducing the practical cost from 5 days on 200 TPUs (Liu et al, 2019b) to less than 3 days on 8 GPUs - thus making the technology much more widely usable.
We are not the first to investigate the combination of KD and NAS, but we are the first proposing a practical approach for doing so.

======  RESULT COMPARISONS

Since our goal is to transfer knowledge from a large model with state-of-the-art performance to a smaller student, we require that the teacher outperforms the student.
As NAGO architectures, used as students, are a strong baseline, we needed to select a stronger teacher in order to make sure there was room for an improvement. Furthermore, it has been shown that better teachers do not necessarily lead to better performance if the student doesn't have the capacity to learn (Mirzadeh et al, 2020).
The purpose of Table 1 is to show how using AutoKD makes it possible to find small models (4.0M parameters) that outperform hand designed students. We couldn't find larger models in the KD literature to compare with more fairly on CIFAR10.

We want to stress that the point is not to say that advanced KD methods are not useful, indeed our approach is orthogonal and could be combined with any of them for even better performance.

[Mirzadeh et al, 2020] Mirzadeh, Seyed Iman, et al. "Improved knowledge distillation via teacher assistant." AAAI Conference on Artificial Intelligence 34(4), 2020.

---

### Author Response · Authors · 2020-11-24
**New version uploaded**

Thanks to reviewer's feedback, we have now uploaded a clarified and improved version of our work. In particular we have added:
- ImageNet results 78.0% top-1 accuracy, an improvement over previous works: 75.5 for Liu et al, 2019b and 76.8 for vanilla NAGO;
- further information on Algorithm 1;
- clarified results in Table 1;
- cleaned up the old Figure 2 (now Figure 8 in the appendix), and clarified the remaining Figures;

More detailed changes, suggested by specific comments, are given in the replies below to each reviewer.

---

### Decision · Program_Chairs · 2021-01-07
**Final Decision**

**Decision:**

Reject

**Comment:**

The paper proposes a new approach to knowledge distillation by searching for a family of student models instead of a specific model. The key idea is that given an optimal family of student models, any model sampled from this family is expected to perform well when trained using knowledge distillation. Overall this is an interesting idea and an important direction of research. However, the reviewers raised several concerns regarding novelty and experimental evaluation. There was a clear consensus among the reviewers that the paper is not yet ready for publication. The specific reasons for rejection include the following: (i) the proposed method is somewhat incremental, and the paper's contributions should be adjusted accordingly; (ii) the experimental results in the paper do not provide a clear/fair comparison with existing approaches, and additional baselines should be considered. The reviewers have provided detailed feedback in their reviews, and we hope that the authors can incorporate this feedback when preparing future revisions of the paper.